# Redesigning metabolism based on orthogonality principles

Aditya Vikram Pandit[1], Shyam Srinivasan[1] & Radhakrishnan Mahadevan[1,2]

Modifications made during metabolic engineering for overproduction of chemicals have network-wide effects on cellular function due to ubiquitous metabolic interactions. These interactions, that make metabolic network structures robust and optimized for cell growth, act to constrain the capability of the cell factory. To overcome these challenges, we explore the idea of an orthogonal network structure that is designed to operate with minimal interaction between chemical production pathways and the components of the network that produce biomass. We show that this orthogonal pathway design approach has significant advantages over contemporary growth-coupled approaches using a case study on succinate production. We find that natural pathways, fundamentally linked to biomass synthesis, are less orthogonal in comparison to synthetic pathways. We suggest that the use of such orthogonal pathways can be highly amenable for dynamic control of metabolism and have other implications for metabolic engineering.

[1] Department of Chemical Engineering and Applied Chemistry, University of Toronto, 200 College Street, Toronto, Ontario, Canada M5S 3E5. [2] Institute of Biomaterials and Biomedical Engineering, University of Toronto, 164 College Street, Toronto, Ontario, Canada M5S 3G9. Correspondence and requests for materials should be addressed to R.M. (email: krishna.mahadevan@utoronto.ca).

It is well established that pathway selection for chemical production plays an important role in metabolic engineering, for example, isopentenol has a higher pathway efficiency on a basis of yield[1] when it is formed from glucose using the 2-C-methyl-D-erythritol 4-phosphate/1-deoxy-D-xylulose 5-phosphate (MEP/DOX) pathway than the mevalonate pathway. Simultaneously, the expression of chemical production pathways is not sufficient to overproduce chemicals because cellular objectives are in competition with a chemical production objective[2]. Therefore, to produce a desired chemical, genetic interventions in the cellular metabolism that couple the growth of the organism to chemical production are seen as necessary. This has been the mainstay philosophy in metabolic engineering, and the literature is abundant with examples of growth-coupled metabolic engineering[3–5].

However, growth-coupled production has many biological challenges owing to the complex nature of metabolism. From an evolutionary perspective, metabolic pathways in cells have been optimized to convert sugar to biomass. The design of these pathways may, partially at least, be explained by optimality principles relating their structure to their function. Examples of such descriptions can be found in literature[4,6–11]. Genetic interventions can change the structure of the underlying metabolic network to force the cell to produce some biomass and some desired chemical. The engineered metabolic network producing the desired chemical has two important characteristics worth noting: (i) It no longer exhibits the optimality principle that is evolutionary in nature and as a consequence, has a lower growth rate compared to the wild type and (ii) the optimality principle that describes biomass production cannot be used analogously to describe chemical production because evolutionary constraints for biomass and chemical synthesis are not the same. Accordingly, since the optimality principle cannot be valid for either chemical or biomass production individually, we suggest that this results in suboptimal production of both.

Since structure is inexorably linked to function, it follows that a network function supporting chemical production and satisfying the key metrics stated above should exhibit a different structure from a wild-type cell and therefore also obey different principles of optimality. For example, in one recent study, the structure of the central metabolism was described as a 'minimal walk' between the input substrate and the 12 requisite precursors for biomass[9]. Hence, based on this minimal-walk description, the natural structure of metabolism is not optimal for the production of a desired chemical.

Here we argue that to optimally convert the input substrate to the target chemical, one has to analogously generate a biosynthetic network that is largely independent of the natural metabolism but still capable of synthesizing biomass. We define such pathways as orthogonal pathways and examine the orthogonal properties of natural and synthetic metabolic networks that are designed for chemical production. This approach is in contrast to the approach taken in contemporary metabolic engineering design where a network optimized for biomass is augmented by deletions. Instead, we seek to determine the design characteristics of an orthogonal network that is optimal for the production of a target chemical as opposed to biomass. In doing so, we also present an algorithmic approach to engineer orthogonal pathways and develop a method based on cut sets to identify metabolic control reactions ('valves') that can be manipulated to allow or disallow cell growth. A metric for optimality that we developed helps to identify pathways that are optimal in the context of a set of minimal cellular interactions.

Our analysis also leads us to consider substrates beyond the sugars that are naturally used by organisms, and to identify substrates such as ethylene glycol that are inherently better suited to produce target chemicals. We show that reworking the metabolic network structure to meet design specifications and designing networks by considering substrate–product pairs has implications for two-stage fermentation design. The tradeoff between growth and metabolite production in such orthogonal strategies can be effectively addressed through the design and identification of enzymes, such as phosphoenolpyruvate carboxykinase, to act as control valves between product and biomass production. Finally, we believe that the approach provides a new paradigm of metabolic engineering strategies for chemicals, in contrast to the existing growth-coupled strategies, which can be difficult to implement in practice. In doing so, it provides an improved framework for industrial strain design and the selection of substrate utilization pathways and strongly suggests that orthogonal strategies can lead to reduced cellular interactions when compared to native pathways that are instead optimized for growth.

## Results

**Defining orthogonal pathways**. Orthogonal pathways are growth-independent pathways optimized for the production of a target chemical. These pathways are characterized by the minimization of interactions between the chemical-producing pathways and the biomass-producing pathways. Physiologically, this means in perfect orthogonal networks (i) the product pathway shares no enzymatic steps with cellular pathways that are responsible for the production of precursors required for biomass and, (ii) only a single metabolite serves as a branch point from which product and biomass pathways diverge. Hence, by design these pathways not only minimize interactions between the cell's biomass-producing pathways and the chemical-producing pathways, but also obey a minimal-walk optimality principle for product formation. The ideal structure of this type of network is shown in Fig. 1a. To enable us to discern between orthogonal and non-orthogonal pathways, we devised a quantitative measure of orthogonality called the orthogonality score (OS) described in Methods section and Fig. 1b,c. A feature of this type of network is its branched pathway structure for biomass and bioproduct formation. The branched structure allows either branch, but specifically the biomass-producing branch to be turned on or off by, for example, controlling the expression of one gene. That enzyme can be called the metabolic valve and its production level can be modulated to attain a desired flux towards biomass. With this theory on orthogonal pathways, first, we examine whether one can observe such orthogonality for the metabolism of sugars such as glucose.

**Natural metabolism is mostly not orthogonal**. Glycolysis, including its many variants, consumes glucose through a highly connected metabolic network. We hypothesized that these points of connectivity, often described as redundancies that make cells robust to perturbations in their environment[12,13], render the native metabolism non-orthogonal towards chemical production. To test our hypothesis, we independently analysed three natural pathways found across the metabolism of cells using a core model of *Escherichia coli*. These pathways, the Embden–Meyerhof–Parnas (EMP) pathway, Entner–Doudoroff (ED) pathway variant and the methylglyoxal (MG) bypass shown in Fig. 2a, are conserved across many heterotrophs[14].

Since analysis of pathways requires a substrate–product pair, we used succinic acid production from these pathways as a case study for our analysis as succinic acid is a well-studied, industrially produced biochemical made from sugars. Using our orthogonality analysis framework (Fig. 1), we show that most natural pathways do not satisfy the principles of orthogonality

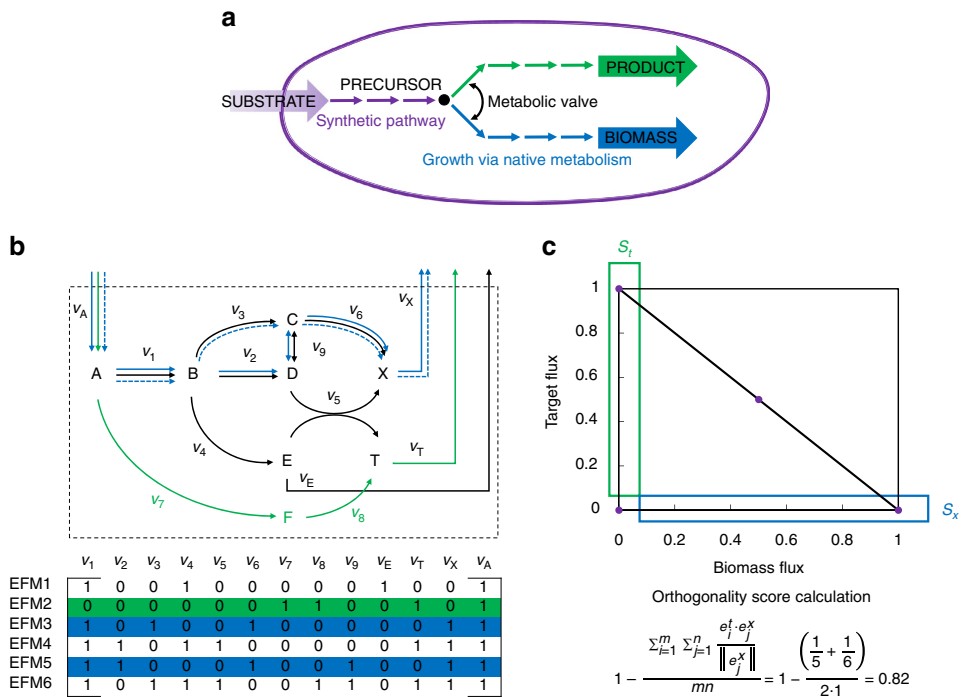

**Figure 1 | The ideal structure of an orthogonal pathway in a cell.** Green corresponds to the EFMs that produce the desired target chemical and are described by the set $S_t$. Blue corresponds to the EMFs that produce biomass and are described by the set $S_x$. (**a**) The branched design is characteristic of this type of orthogonal structure. (**b**) We show a hypothetical small network where A is converted to products E, X (biomass) and T (target compound). The mathematical representation of this network is described by the EFMs shown below the network in a Boolean matrix, where blue lines are the biomass-only forming EFMs (3 and 5) and green is the product-only forming EFM (2). This type of network structure can be described as an orthogonal network because A can be converted to T by reactions $v_7$ and $v_8$ and the metabolic valve $v_1$ can be modulated to be turned on or off. Traditional metabolic engineering strategies would attempt to drive flux towards the desired product, T, by growth coupling T to X. For example, this may require the deletion of $v_3$, $v_6$ and/or $v_7$. Orthogonal metabolic-engineered strategy relies on the thermodynamics for converting A to T and manipulating $v_1$ to control flux towards biomass. An example calculation of the orthogonality score is shown. (**c**) We show the production envelope for the network containing the EFMs that describe that solution space. The functionalities of interest of the network are shown in the green and blue boxes. These represent the desired subspaces $S_x$ containing the elementary modes $e_j^x$(EFM3, EFM5 shown in blue) and $S_t$ containing the modes $e_i^t$ (EFM2 shown in green). The orthogonality score is calculated based on the similarity of these subspaces.

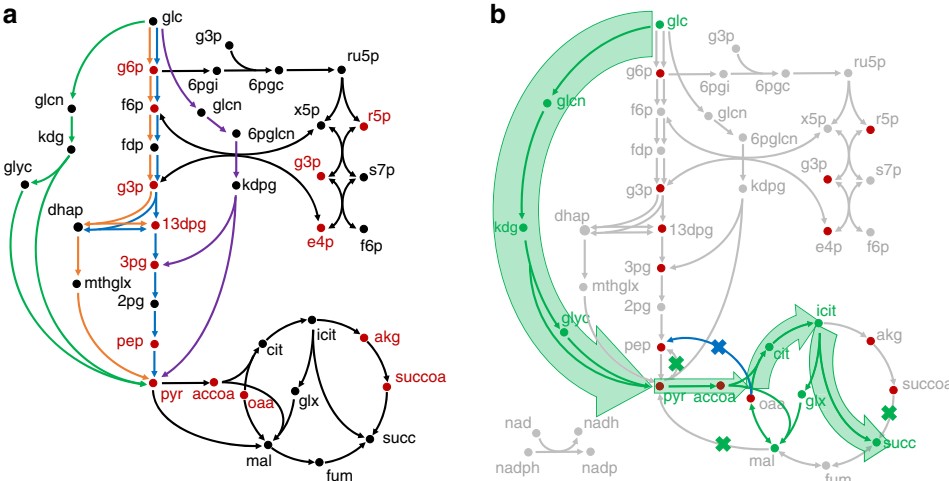

**Figure 2 | Simplified metabolic map of the glucose consuming pathways.** (**a**) The unique pathways analysed in this study are indicated by the coloured lines. Green: glucose synthetic; blue: glycolytic EMP; orange: MG bypass; purple: ED pathway. (**b**) Sample cut set strategy for synthetic glucose pathway shows that the structure is amenable to a metabolic valve topology which bypasses most of the biomass precursors. These precursors of the central metabolism are required for growth and have been identified in red. The green x marks which reactions have been identified for deletion by the algorithm to design for orthogonality. The blue x marks the metabolite valve. Synthetic pathways attempt to bypass these precursors as well as the points of regulation. A similar branched topology was not observed for natural glycolytic pathways.

for the production of succinic acid from glucose. Briefly, the orthogonality score provides a quantitative measure of the ability of the metabolic network to support two distinct objectives. A value of 1 signifies that biochemical production is essentially orthogonal to native metabolic network and can be described as a biotransformation while a value closer to 0 means that there is a significant overlap with the biomass-producing network (see Methods section and Supplementary Fig. 1). Hence, larger values are indicative of a more orthogonal network, implying that the separation of biomass and product-producing reactions should, theoretically, be easier to achieve. In the present case, the competing objectives are the production of biomass and the production of succinate from glucose.

The orthogonality score for succinate production for each of the natural pathways is shown in Table 1. Orthogonality scores for these natural pathways range from 0.41 to 0.45. We then analysed a synthetic pathway for glucose utilization and its conversion to succinate, and compared it with the natural counterparts identified earlier. The pathway was identified using a pathway predictor algorithm similar to those in the literature[15], but interestingly has also been suggested for cell-free applications[16]. Figure 2a shows the synthetic glucose pathway, which bypasses glucose phosphorylation and all biomass precursors of glycolysis to directly produce two moles of pyruvate. Pyruvate is then carboxylated to oxaloacetate and follows the typical reductive or oxidative branches of the tricarboxylic acid (TCA) cycle. In contrast to orthogonality scores for the natural pathways (0.41–0.45), this synthetic pathway has a larger orthogonality score, 0.56, than any of the natural EMP, ED and MG pathways.

We find that, within the natural pathways, the difference in orthogonality arises from the degree to which the glucose utilization pathways overlap with elements of the metabolism that support biomass. Both the MG shunt and the less connected ED provide routes that bypass several biomass precursors. This can be determined by analysing the elementary flux modes (EFMs) that only produce the target chemical (set $S_t$, Fig. 1c) and calculating the total number of reactions having a non-zero flux through a biomass precursor metabolite across all EFMs (Table 1). Both exhibit a higher orthogonality score and a lower average number of precursor forming reactions per EFM. These results are in general agreement with the principles that the orthogonality score metric seeks to capture.

Where possible, the orthogonality scores of Table 1 were compared using Kolmogorov–Smirnov test against the EMP distribution to verify that the mean comes from different distributions. The test, when applied to the synthetic glucose pathways, showed a difference in their underlying distributions. Hence, by using the wild-type glucose network as a threshold, we can interpret these scores as a qualitative measure of the degree to which substrate utilization pathways will result in chemical production in a way that is more or less independent of growth to

the highly connected wild-type core network. In other words, core networks with orthogonality scores greater than $\approx 0.5$ begin to exhibit dissimilarities in properties and structure between target chemical and biomass pathways.

A second observation was that orthogonality and redundancy are negatively correlated. Since orthogonality quantifies the shared nature of the biomass and the chemical-producing pathways within a metabolic network, we find that increasing the redundancy of the network decreases its underlying orthogonality. As an example, the inclusion of phosphofructokinase, which is typically unique to the EMP pathway, in the network of the ED pathway, reduces the orthogonality score (0.43). Hence, increasing the number of redundant reactions common to product and biomass synthesis decreases the orthogonality of the two objective functions. This result is expected as the goal of orthogonal networks was to share the least number of reactions and thereby reduce redundancies in the network. Hence, as hypothesized, by eliminating shared redundant pathways between the product and biomass precursors, well-designed synthetic pathways can reduce the complexity of supporting two distinct production objectives relative to the wild-type network. Next, given the popularity of growth-coupled strategies, we analysed the impact of growth-coupled strain design on the orthogonality between production and growth.

**Growth-coupled strategies are not orthogonal**. In growth-coupled strain design, cell growth is linked to product formation by identifying reactions such that biomass-producing EFMs (set $S_x$ in Fig. 1b in blue) are removed. When all EFMs are removed from this set, the strain can be said to be strongly coupled[17]. Since no EFMs are left that produce biomass without producing the target chemical (in set $S_x$), the orthogonality score is undefined (as there are no biomass production modes left) and it can no longer be calculated using equation (1).

However, in certain cases, it is possible to couple chemical production to growth without removing all EFMs in $S_x$. This scenario is known as weak coupling[17], and the orthogonality score can be calculated. Growth coupling strategies that are weakly coupled have a score similar to or lower than the wild-type network (Table 2). Hence, we find it is not possible to minimize interactions within natural (non-orthogonal) networks by growth coupling to obtain orthogonality. This can be explained because the goal of these methods is to couple two divergent objectives, and not to enhance the orthogonality. In contrast, the use of synthetic pathways transformed into the host organism can bring about orthogonality in the metabolism of cell factories. As a natural progression of these results, we compared the orthogonality score of the synthetic pathways for succinate production with natural pathways.

**Orthogonality is greatest for branched structures**. In the introduction, we described that ideal orthogonal pathways should

---

**Table 1 | Summary of orthogonality calculations.**

|  | Substrate utilization pathway | | | | | | |
|---|---|---|---|---|---|---|---|
|  | **EMP** | **ED** | **MG** | **Natural xylose** | **Synthetic glucose** | **Weimberg** | **Synthetic MEG** |
| Score | 0.41 | 0.45 | 0.43 | 0.36 | 0.56 | 0.57 | 0.62 |
| Total precursor supporting reactions | 82,236 | 67,059 | 176,575 | 86,499 | 3,610 | 2,233 | 464 |
| Average precursor reactions/EFM | 11.2 | 8.6 | 10.5 | 12.8 | 6.3 | 6.6 | 3.3 |

The orthogonality scores for the various pathways either synthetic or natural consuming glucose, xylose or ethylene glycol and producing succinic acid are shown. These scores are calculated from the EFMs of the *E. coli* core model, using equations (1) and (2). The model was modified as necessary to include the reactions for each pathway. The total precursor supporting reactions correspond to the total number of reactions that produce one of the 12 precursor metabolites and is active in each mode, across all EFMs belonging to the space $S_t$. They correspond to the intersection that chemical production has with biomass formation. The orthogonality score implicitly accounts for this intersection, and the underlying negative correlation is reflective of the relationship between biomass production and orthogonality.

**Table 2 | Summary of orthogonality scores for two types of networks.**

| Substrate utilization pathway | Growth coupled | Orthogonal network designed by genetic interventions | | | |
| --- | --- | --- | --- | --- | --- |
| | EMP | EMP | Synthetic glucose | Weimberg | Synthetic MEG |
| Orthogonality score | 0.39 | 0.41 | 0.55 | 0.56 | 0.62 |
| Number of biomass precursors synthesized | 12 | 11 | 5 | 2 | 1 |
| Protein cost and thermodynamic contribution (g s mol$^{-1}$) | — | 12 (7%) | 3,600 (0%) | 110 (0%) | 47 (6%) |

The growth-coupled score occurs for a set of gene deletions that couple biomass growth above 0.05 h$^{-1}$ and product yield >1 mol per mol. The orthogonal network designed by genetic interventions scores are calculated after applying the ValveFind algorithm described in this publication. The score is calculated for a reduced network after removing reactions in the cut set, but leaving the valve reaction in the on position. The table also shows the total number of biomass precursors that can be formed when the metabolic valve is closed. The cost of operating the pathway is provided using 10 mmol gDW$^{-1}$h$^{-1}$ as a basis for the calculation. The values represent total protein cost and the contribution of the thermodynamic cost is shown in parenthesis.

have branched structures. This type of topology is valuable because it permits chemical production to be separated from biomass production by a nearly independent subsystem that is modular and distinct from the rest of the metabolism. We found that this type of independence is expectedly absent in natural metabolism, but could be engineered by the use of synthetic pathways, and be numerically quantified by the orthogonality score. Our present discussion extends these results by studying branched network structures in metabolism. In this section, we ask whether these types of topologies can be found in natural metabolism or whether they are a characteristic of orthogonal pathways for substrate utilization. To answer this question, we exploit the property of minimal cut sets (MCS) that eliminates the biomass production pathways and enforces production above a threshold yield.

The MCS algorithm allows for the identification of designs with a non-zero production of the target chemical that lead to orthogonality by removing all the growth-dependent pathways (whether coupled or not), resulting in zero growth. We use this algorithm and develop a novel approach ('ValveFind') to identify valve reactions that permit orthogonal pathway design. Specifically, the MCS algorithm is used to search for cut sets that guarantee theoretically viable product yields when the growth rate is zero. By applying these cut sets to the metabolic network, and then searching for reactions in the cut set that, when active, can restore growth rates above a desired threshold (for example, 90% of the wild-type growth rate), it is possible to identify metabolic valves. If the reaction can restore biomass growth, then the cut set can be considered as a candidate for a branched structure. By calculating its orthogonality score when the valve is considered on, permitting flux (if the valve is off, growth is not possible), the cut sets' suitability can be assessed from the perspective of orthogonality. This approach necessitates a dynamic metabolic engineering (DME) strategy consisting of a growth phase and a subsequent production phase. We provide a detailed explanation of the approach in Methods section.

First, since earlier results indicated that glycolytic pathways for consuming glucose and producing biomass were not orthogonal to succinic acid production pathways, we tested the ValveFind methodology on the natural glycolytic metabolism. In this case, we sought to determine whether a branched structure for succinate production could be derived through a set of gene deletions in a network characterized by a low orthogonality score, thereby effectively raising its score. This is akin to the method used to obtain strain designs for metabolite production albeit without demanding growth as is common in these methods. We identified 99 cut set strategies capable of producing succinic acid independent of growth. Of these, only 38 contained a reaction that could serve as a metabolic valve not linked to nutrient limitation (for example, ammonium uptake). TCA cycle reactions such as isocitrate dehydrogenase and fumarase reactions were among the most commonly identified valve reactions. An example of one of those designs is the deletion set encompassing

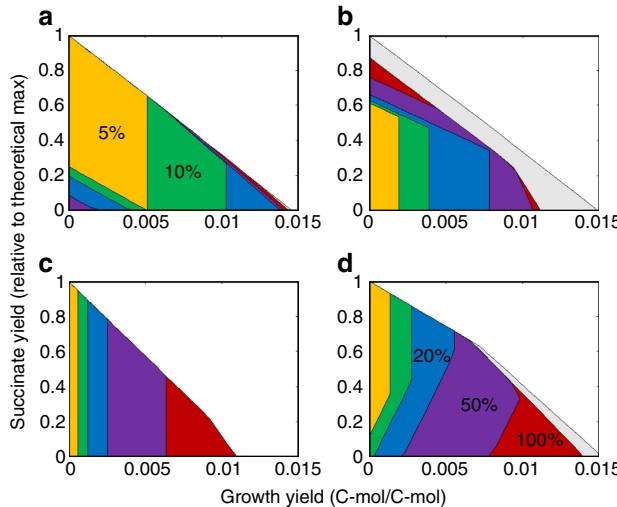

**Figure 3 | Production envelope for succinate production.** (**a**) Glucose utilization by glycolysis, (**b**) glucose utilization by the synthetic pathway, (**c**) xylose utilization by the pentose phosphate pathway and (**d**) xylose utilization by the heterologous synthetic Weimberg pathway. These envelopes capture the solution space. By controlling a single reaction, it is possible to shrink the solution space to a smaller defined region of higher product flux. Grey indicates the unmodified network. The metabolic valve is then modulated from 100% open (red) to 50% (purple), 20% (blue), 10% (green) and 5% open (yellow).

reactions phosphoenolpyruvate carboxykinase, a transketolase and both malic enzymes. Restoring isocitrate dehydrogenase could restore growth above 90% of wild type (Fig. 3). This sample cut set has an orthogonality score of 0.41.

We then calculated the orthogonality scores for all 38 sets and manually verified that none exhibited an obvious branched structure. We initially expected all cut sets to be lower than the wild-type score of 0.41 since we expected that branched structures would be absent from the natural metabolism. Instead, we found that the scores varied considerably between cut sets, and the maximum was 0.43 and the minimum was 0.28. In hindsight, it is intuitive that orthogonality scores can be both greater and less than the unmodified network score. Removing reactions that contribute to redundancy in biomass space increases the orthogonality score if those reactions support biomass synthesis, as shown earlier. In contrast, if those reactions disproportionately remove EFMs that support product formation, it is possible for the score to decrease. Hence, the results suggest that cut set-based design strategies can be selected rationally to minimize the interactions between biomass and chemical production EFMs, even in natural metabolism where branched structures are not apparent. Low scores can be disregarded as they are not suitable for the primary objective of orthogonal metabolism, which is the minimization of interactions.

High scores, however, do not necessarily guarantee branched structures, although branched structures, as we will describe below, do result in high scores. In the example, controlling isocitrate dehydrogenase as a metabolic valve prevents cell growth, but a zero flux through that reaction does not preclude the synthesis of most of the individual component metabolites of biomass. Specifically, synthesis of 11 of the 12 biomass precursors is possible even when biomass as a whole cannot be synthesized. The orthogonality score captures this dependence that the individual components of biomass have on network interactions, which can be indiscernible by simply examining individual valve reactions. The relatively lower-orthogonality score for this case (0.41) compared to the synthetic pathways in Table 2 captures the dependence that biomass-only EFMs have on chemical production. Hence, a metabolic valve that can restore growth to wild-type levels is not suitable as the sole criteria for orthogonality.

Next, we applied this algorithm to the synthetic glucose pathway. We found 131 out of 367 cut set strategies satisfying the condition that there is at least one metabolic valve reaction that can restore growth to 90% of wild type. Consider one sample strategy from that set of 131, which requires the disruption of five genes in addition to the standard fermentative enzymes (*ldhA*, *adhE*, *ackA*). These five consist of the gluconeogenic enzymes phosphoenolpyruvate synthase, both malic enzymes as well as succinyl-coa ligase (Fig. 2b). Under this strategy, these four gene deletions are required to provide a singular pathway towards satisfying growth precursor requirements. The fifth genetic intervention, phosphoenolpyruvate carboxykinase, is the metabolic valve that can be manipulated to direct flux towards biomass to chemical synthesis. When the metabolic valve is off, only five biomass precursors that also belong to the succinic acid biosynthesis pathway can be synthesized, and the mean orthogonality score is 0.55, a relatively high value. Hence, this combination of genetic interventions allows the metabolism to be recast into a design with one metabolic valve.

Hence, branch-like topologies, which could not be found in natural metabolism for succinate production from glucose, were a characteristic of synthetic pathways for substrate utilization. Additional designs obtained by ValveFind are provided in the Supplementary Data 1.

**Metabolic valves efficiently reduce the solution space.** Figure 3 shows that for every design obtained through the cut set analysis, the metabolic valve reduces the solution space efficiently to a small production envelope, and if possible, a single EFM. The area of each envelope is representative of the effectiveness of separating biomass from chemical production. It can be seen that, for the case of the native glycolytic pathway, the flux through the valve needs to be reduced to <5% of the wild-type value before the biomass yield is reduced significantly ($<0.005\,\mathrm{mol\,mol^{-1}}$). Whereas, for the synthetic pathways, the same reduction in yield can be achieved with a valve flux that is 10–25% of the wild-type flux. These results further highlight the value of orthogonal design for modulating flux and the differences in the orthogonality between the native and the synthetic substrate utilization pathways. We extend this analysis to the xylose metabolism of *E. coli* as well to show that these principles appear to be consistent (Fig. 3).

**Orthogonality depends on the substrate utilization pathways.** Metabolic pathways are inherently dependent on the input substrate. We explored the role of substrate selection on achieving orthogonality. First, we examined xylose as a substrate for succinate production. An analysis performed for xylose showed

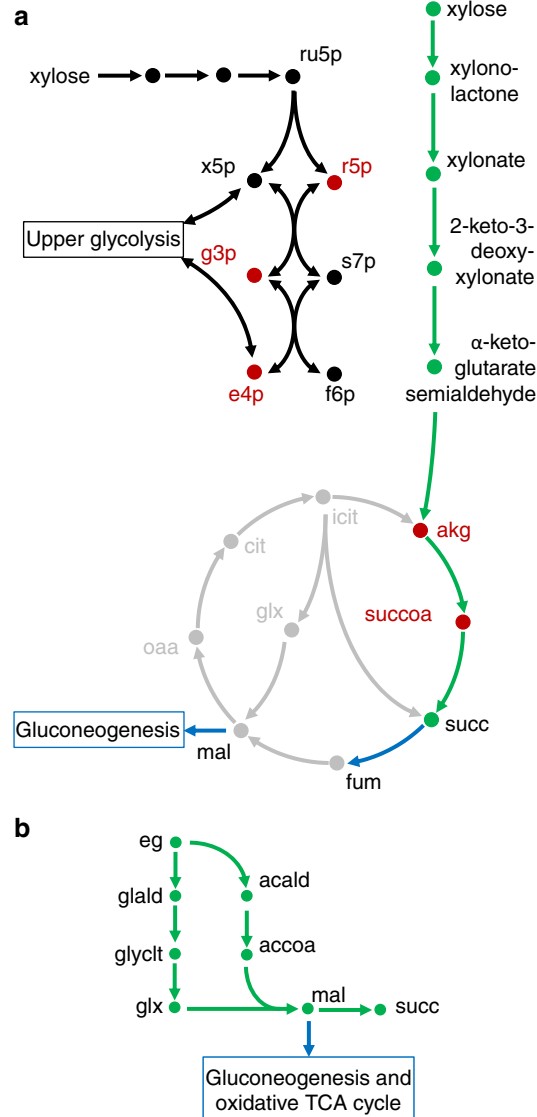

**Figure 4 | Orthogonal pathway design for other substrates considered in this study.** (**a**) The Weimberg pathway is heterologous to *E. coli*, however, it provides an efficient route for xylose assimilation that bypasses the central carbon metabolism and most biomass precursor molecules. To the left of the Weimberg pathway is shown the natural route for xylose assimilation in *E. coli* through the pentose phosphate pathway. Succinate dehydrogenase, which converts succinate to fumarate is an ideal candidate as a metabolic valve (shown in blue) as it allows flux to the TCA cycle and supports gluconeogenic pathways for cell growth. (**b**) The orthogonal routes for ethylene glycol assimilation examined in this study. Malic enzyme is an ideal candidate for a metabolic valve (shown in blue) as malate decarboxylation to pyruvate can support cell growth. The degree to which the pathway overlaps with the central carbon metabolism is captured by the orthogonality score for each specific pathway.

that the native pathway of xylose utilization in *E. coli*, which is assimilated by the pentose phosphate pathway was, as expected, not orthogonal (score: 0.36). However, conversion of xylose to succinic acid was highly orthogonal for the non-native Weimberg pathway (Table 1 and Fig. 4a). Once again, these results suggest that the type of xylose utilization pathway has a significant impact on orthogonality. This result has been borne out by the recent study on the use of this pathway for the production of 1,4-butanediol without major metabolic engineering of the

 

native metabolism of the cell[18]. As a natural progression, we asked what other substrates can be suited for succinate production? We looked at a variety of non-traditional substrates that could be derived from $CO_2$, and found that ethylene glycol is an excellent substrate for orthogonal metabolism.

Ethylene glycol enters the metabolism at the malate node and proceeds to succinate by the reductive branch of the TCA cycle. A well-designed orthogonal pathway (Fig. 4b) has a much larger orthogonality score than glucose (Table 1), and its metabolic valve can restore growth rate to 90% of the wild-type rate. Malic enzyme acts as a control valve for the network. These results have two important implications for industrial biotechnology. The first is that, while glucose is a natural substrate for microbes, it may not be the best for chemical production. Therefore, it is important to consider how unconventional feedstocks, especially those that can be derived from $CO_2$, as in the case for ethylene glycol, can be used in biological processes to optimally produce a desired chemical. The second related consequence is that the substrate utilization pathway is an exceptionally important criteria for orthogonal design.

Reflecting on these results, we find that the substrate utilization pathways determine orthogonality primarily in two ways: (1) it provides non-phosphorylated routes to assimilation, which bypass regulation in the metabolism and, (2) it allows the substrate's entry point into the metabolism in a way that bypasses highly connected nodes of natural metabolism. Both xylose to succinate and ethylene glycol to succinate are examples of these types of pairings. For example, in the context of orthogonality, glucose conversion is not as well suited to succinate production compared to ethylene glycol, which has a substantially higher score. In addition, we wanted to evaluate whether the orthogonality results obtained for succinate production could be extended to other products. Hence, we examined a total of nine different pathways and five additional products (adipic acid, 1,4-butanediol, 2,3-butanediol, ethanol and isobutanol) from four substrates to support the generality of the findings in this case study. See Supplementary Table 1 for these additional case studies. These results clearly suggest that our findings are not specific to succinate production alone and can be generalized. Moreover, we find some interesting cases, such as glycerol conversion to 2,3-butanediol or 3-hydroxypropionic acid (Supplementary Note 2), that reveal how natural metabolism can, under certain substrate–product pairings, be regarded as orthogonal. Finally, we wanted to understand the potential tradeoffs that might occur during orthogonal design when traditional metabolic pathways optimized for growth are bypassed to maximize orthogonality. We hypothesized that one such tradeoff might involve the protein cost associated with these synthetic pathways relative to the native pathways. Hence, we investigated these costs using the framework presented in Flamholz et al.[9] for comparing the enzymes costs for the different glycolytic pathways in E. coli.

**Cut set design allows calculation of pathway energetics.** Flux through metabolic pathways for biomass and product synthesis are determined by, among many factors, hierarchical cellular regulation that favours biomass synthesis over product synthesis. However, this flux is also a function of the driving force available through that pathway, expressed as changes in the Gibbs free energy and the kinetic parameters of the enzymes in the pathway. Coupling product formation to growth overrides the cell's regulation, that in the presence of a driving force, permits product synthesis and provides a basis for rational engineering to increase that flux.

Since orthogonal pathways exist outside any regulatory framework, only a driving force is required to support flux

through them. By examining the energetics of their path, we can understand how cell factories can support flux through these pathways. Hence, we calculated the various components of the protein cost: the kinetic, thermodynamic and saturation cost[19,20].

We calculated the protein costs of the synthetic pathways and the natural glycolytic pathway (Table 2). The protein costs of the pathways varied from 12 to 3,600 g s mol$^{-1}$. The synthetic glucose pathway was the most expensive—two orders of magnitude greater than the natural EMP pathway. This difference occurs as a result of poor kinetics of a single enzyme that dehydrates glycerate to pyruvate. In general, these results show that the difference in the cost of supporting flux can vary depending on the pathway. However, not all orthogonal pathways have high protein costs. For example, the orthogonal ethylene glycol pathway has costs of 47 g s mol$^{-1}$ and the Weimberg pathway has a cost of 110 g s mol$^{-1}$.

We find an interesting observation when looking at the three components that make up the protein cost. A feature of orthogonal pathways is that they involve non-phosphorylative reaction steps that are orthogonal to the phosphorylative steps that are typically found in the metabolism of biomass pathways. The synthetic pathways have an overall higher cost because the thermodynamic advantage from non-phosphorylating reactions is detracted by a higher kinetic penalty for using inefficient enzymes. For example, the synthetic pathway has almost no thermodynamic penalty while the glycolytic pathway has a 10% penalty. This suggests the possibility that successful enzyme engineering or screening of non-phosphorylating enzymes with better kinetic parameters might lead to orthogonal pathways capable of supporting a higher flux than natural pathways due to their thermodynamic advantage. Taken together, these results reasonably suggest that cell factories that utilize non-natural pathways for substrate utilization may be able to more efficiently support flux for chemical production.

**Discussion**

In this paper, we provide an alternative perspective to the problem of designing pathways and strains for metabolic engineering. In contrast to the prevalent approach of growth-coupled designs, we suggest that orthogonal pathway design coupled with DME might be effective for de novo strain design. This idea of orthogonality is closely related to modularity, which has been well studied for metabolic networks[21–23], and used for metabolic engineering[24–26]. While, the central metabolism of E. coli is highly connected and robust, elements of it do behave as modular subsystems. Amino-acid biosynthesis control is one such example that allows the cells to be stable in the presence of varying environmental conditions[27]. Regulation at the beginning and end of these subsystems allows cells a control mechanism well suited for robust growth. Orthogonality principles can be thought of as modular subsystems for chemical production that minimize total interactions with the natural cellular metabolism, and that can be achieved through synthetic pathways for substrate utilization.

When traditional metabolic engineering aims to repurpose cellular metabolism for chemical production, it does so within the evolutionary disposition for growth known as growth coupling. However, the organization of this network structure follows principles of optimality different from those that metabolic engineers would attribute to be optimal for chemical production. We have shown efficient chemical production requires an optimality principle outside the scope of a cellular growth objective, which, akin to elements of metabolism such as amino-acid biosynthesis, require modular and independent subsystems in the cell, and a robust control mechanism over them. In this

 

work, these subsystems can be measured by the ability of the metabolic network to perform two separate tasks (growth and chemical production). The orthogonality score measures this ability by calculating a 'distance' metric in the metabolic flux space for these two tasks.

A determinant of orthogonality is the overlap of the reactions that support biomass production and the chemical production pathways. A key finding of our work is that native glucose utilization pathways are not orthogonal for succinate and several other products (for example, 1,4-butanediol) due to this overlap. Further analysis reveals that this non-orthogonality is largely due to the generation of phosphorylated metabolites and the individual biomass precursor metabolites in these native pathways that are valuable for biomass production, but are not essential for substrate utilization in the chemical production modes. In the Supplementary Note 2, we expand on several additional case studies that support these findings.

We found that, by contrast, the catabolism of most orthogonal pathways lacked phosphorylation reactions. We found both glucose and xylose to be structurally more efficient for product formation when they were not phosphorylated. These types of non-phosphorylated pathways are sometimes observed naturally in microbes, although they are not common. These pathways typically do not involve substrate-level phosphorylation, are less energy-efficient and dissipate more free energy, thereby providing a higher thermodynamic driving force than conventional pathways. This is an important aspect of the flux capacity of metabolic pathways.

There are two significant benefits for bypassing biomass precursors: (1) Pathways produce higher yields because they avoid carbon losses associated with precursor synthesis. For example, the generation of metabolites of the pentose phosphate pathway results in carbon loss through *zwf*. (2) Orthogonal pathways implicitly bypass regulation as the biomass precursors tend to be highly regulated. For example, fructose-1-6-bisphosphate has been demonstrated to be a metabolic 'flux sensor' important to the control of glycolytic flux[28]. Other such metabolites also act to regulate the cell, and changes in their concentration have ripple effects through several metabolic pathways. Hence, synthetic orthogonal pathways offer a metabolic solution to a complicated regulatory problem.

The significance of a flux sensor in natural metabolism is an important consequence for metabolic engineering. Glycolytic flux during stationary phase often ceases due to the accumulation or draining of intracellular metabolites, which are recognized by these flux sensors, and play a role in reducing glycolytic flux[28]. Hence, most chemical production in industry is carried out using a fed-batch process, where the goal is to engineer a high glycolytic flux during stationary phase by targeting the regulatory network[29]. Orthogonal pathways rely on these same principles of using a thermodynamic driving force for conversion, but avoid the necessary challenges of targeting regulatory networks.

We also uncovered that orthogonality principles rest on the pairing of an input substrate and the product. Accordingly, engineering pathways *de novo* for a given substrate–product pair is a better approach to metabolic engineering than depending on pathways that consume glucose for any and all biochemical products. The diversification of feedstocks away from glucose, syngas, methane, methanol and glycerol, supports our idea[30–34]. Our framework applies principles of orthogonality to design metabolic processes that are tailored for the conversion of a specific substrate to a product in the most efficient way possible.

Our work also has important applications for DME. Conceptually, DME has gained quite a bit of attention[35,36], and shown early promise[37–44]. Several studies have utilized strategies for controlling pathway flux to improve yields using inducible

systems and circuits, as well as metabolic sensors connected to synthetic cell circuits[39,45]. However, adapting these early successes to high yielding industrial strains has yet to be shown. The balancing of gene expression in DME, through multi-gene control, is among the many challenges. In a typical highly regulated network, this requires global coordination of metabolism. Studies employing the use of synthetic circuits to control several genes seem to be limited over the number of genes they are capable of accurately controlling. Our analysis suggests that orthogonal pathway design may be a key to experimentally realizing this in industrial strains. The orthogonal design proposed here reduces the number of interactions within metabolism and facilitates a two-stage fermentation strategy. It achieves the goal of circumventing the complex regulatory, enzymatic and metabolomic changes by controlling the flux towards biomass precursors via a metabolic control valve. Importantly, two-stage fermentation (or growth-uncoupled production) is typically used in commercial bioprocesses for large-scale chemical production despite the fact that so many strain design algorithms are focused on growth coupling. In this regard, our framework provides a direct route to translate lab-scale designs to commercial strains without first developing growth-coupled strains that are not suited for two-stage industrial production.

It is worthwhile noting that nutrient-based valves can exist, and there have been demonstrations of such valves including the use of oxygen[46], nitrogen[47] and phosphate[48] limitations. For instance, oxygen-based nutrient valves have been observed in two-stage fermentation for succinic acid production, and nitrogen limitation has been used to produce citrate. However, computational strain design and early strain development has conventionally been guided by a growth-coupled approach. Hence, we have proposed a computational approach to design valves for DME and suggest that future research could focus on even more efficient methods for the design of metabolic control valves based on orthogonal pathways.

The recent focus in metabolic engineering has been the design and use of complex synthetic circuits to control gene expression (for example, via a synthetic toggle switch[42,49]). In light of these approaches, our work has been to understand how reworking the design of the central metabolism may allow the simplification of these circuits, so that rather than employing a multi-gene control, it may be possible to achieve the desired production target(s) by manipulating a single gene. Of course, it is conceivable that these gene-level valves could be combined with the valves related to nutrient uptake to provide an additional layer of flexibility in controlling metabolism.

Finally, implementation of synthetic substrate utilization pathways is not common. However, a growing body of successful experimental studies supports the value of such synthetic pathways[50–52]. This strategy has been recently applied for the design of a synthetic ED pathway in *E. coli*[12]. Our approach formalizes the advantages of such synthetic pathways, and provides a systematic framework for introducing synthetic orthogonal pathways for metabolic engineering.

One of the many challenges that we do not explicitly consider in our current analysis are protein-level interactions of orthogonal pathways. These include enzyme-level inhibition by cofactors or cellular metabolites. The issue of promiscuity of enzymes within metabolism is also another issue that needs consideration. Nevertheless, these are issues that are currently confronted and addressed by almost any metabolic engineering design approach during the scale-up of high-yield strains. Hence, these issues are not a new task for metabolic engineers.

Most importantly, to our knowledge, this work represents the first time that the role that substrate utilization has on metabolic

engineering and chemical production has been evaluated, outside of pathway yield. In the introduction, we had noted that cellular metabolism has been shaped by evolutionary forces for cell growth and survival, objectives which are at odds with chemical production. To understand how 'far' apart metabolism is between growth and chemical production, we have proposed a mathematical framework for systematically evaluating this distance. In some cases, chemical production can be satisfactorily obtained by natural pathways, but more often it is useful to engineer synthetic pathways for substrate utilization.

In conclusion, we derive principles for metabolite production using pathways that interact as little as possible with the cell's natural metabolism. Taken together, we believe our work bridges the current methodologies of strain design at the lab scale to the design of industrial growth-independent production strains that are necessary to satisfy key fermentation metrics that make bio-production a financially viable process[53,54]. The development of industrial microbial strains typically focuses on improving flux through the central metabolism under the assumption that efficient growth pathways are also valid for product synthesis. Studies have shown that more efficient chemical production can be achieved when heterologous enzymes are engineered into the cell to bypass certain biomass precursors. Our work extends these circumstantial observations into a formal mathematical framework and shows that full pathways that avoid many biomass precursors can produce chemicals through optimal network structures.

## Methods

**Definitions.** Briefly, orthogonality refers to the ability of a metabolic network to support optimal metabolite production independent of growth. The ideal orthogonal network is characterized by the presence of at-most two independent branches coming out of a common node (a metabolite). Each branch should contain reactions entirely devoted to the production of either the product or biomass. The common metabolite serves as an intermediate compound from which biomass precursors as well as the desired biochemical can be produced in the two different branches. We provide a metric, the orthogonality score, that quantifies the network's ability to convert a substrate to a product with as few shared nodes as possible, between the reactions that are responsible for producing biomass and those that are responsible for the conversion of the substrate to the product.

**Measuring orthogonality by a metric.** If the stoichiometric solution subspace of reactions contributing to product production and biomass production can be represented by $S_t$ and $S_x$, respectively, the orthogonality score measures the degree of separation of $S_x$ and $S_t$. It also captures the complexity of moving between the $S_t$ and $S_x$ subspaces as a function of the average number of reactions that need to be turned 'on' or 'off' in EFMs to move between subspaces. This measurement is akin to the Euclidean Norm measuring the distance between any two points in $n$-dimensional space. Geometrically, the score characterizes the complexity of separating the reactions that contribute to product production from the reactions that contribute to biomass formation. Accordingly, by measuring the average similarity (AS) between two shared parts of the same network, orthogonality enables one to make decisions regarding the ability to uncouple biomass from product production.

The calculation of the orthogonality score uses EFMs[55,56]. Once EFMs of a given network are enumerated, we split them into two distinct sets corresponding to $S_x$ and $S_t$ containing their respective EFMS, $e_j^x$ and $e_i^t$. $e_j^x$ is determined by those EFMs that contain a non-zero flux through the biomass reaction but not through the target chemical flux, while $e_i^t$ EFMs contain a non-zero flux through the target product reaction but have zero biomass flux. The score is calculated from the AS coefficient of the reactions that are common to supporting only EFMs that produce biomass ($e_j^x$) and only those that produce the target compound ($e_i^t$), and normalized to the size of the biomass supporting network.

$$\overline{AS} = \frac{\sum_{i=1}^{m} \sum_{j=1}^{n} \frac{e_i^t \cdot e_j^x}{e_j^x \cdot e_j^x}}{mn} \quad (1)$$

$$OS = 1 - \overline{AS} \quad (2)$$

The dimension of the dot product calculated as the AS of the EFMs of the sets $S_x$ and $S_t$ (equation 1) quantifies the number of shared reactions between the subspaces divided by the total modes, $m$, in the set $S_t$ and the total number of modes, $n$, in $S_x$. A large orthogonality score is obtained when many reactions are shared between the two subspaces, and a smaller orthogonality score indicates a

greater degree of separation between reactions contributing to $S_x$ and $S_t$. In cases where the underlying distribution of the orthogonality score was from a bimodal or multimodal distribution, as determined by the bimodality coefficient[57], the highest mode was taken as the orthogonality score for the network.

In addition to quantifying the degree of orthogonality of any given natural or synthetic metabolic network, it is also possible to design and construct pathways that are orthogonal, and rank their orthogonality on the basis of the aforementioned score. We describe a methodology to achieve such orthogonal pathways as a novel application of MCS typically used in *in silico* strain design.

**Determining MCS and control reactions.** The ValveFind algorithm identifies a set of interventions and a candidate metabolic valve reaction in the network evaluated as a function of the minimization of the average interactions between chemical production and biomass production. The deletions serve to funnel all the carbon flux through for product production and the metabolic valve helps to identify network structures that may be amenable to a branched topology.

In this work, ValveFind uses the core model of *E. coli*[58] and the MCS algorithm available as part of *CellNetAnalyzer*. The MCS algorithm primarily uses a mixed integer linear programme to solve for MCS[59,60]. To identify the set of genetic interventions, we search for MCS reactions that are required to be removed to guarantee a yield greater than a desired product-yield threshold without demanding growth. This method retains EFMs on the $\mu = 0$ hyperplane above a yield threshold, and may also retain EFMs contained within the production envelope, which are growth coupled. Then, the ValveFind algorithm ranks cut set designs by their orthogonality score by exploiting our theory on the overlap of EFMs. Efficient designs that reduce EFMs in the hyperplane, to a single point if possible (depending on the threshold product yield demanded[17]), will have a high score. However, those designs that retain growth-coupled EFMs present in a small high-yield region of the production envelope will have a lower score.

For the sugar substrates in this study, we used 1 mol per mol as the minimal yield threshold. Substrate uptake reactions were considered at 20 mmol gDW$^{-1}$ h$^{-1}$. Due to the dependency of the cut sets on initial exchange reactions permitted in the model, these were included or excluded on a case-by-case basis, and is described further in the sample results in the Supplementary Data 1 and Code files.

With MCS identified, the ValveFind algorithm applies flux variability analysis on each reaction in each cut set to determine the ability of every reaction within every cut set to restore growth. Within a given cut set, a reaction that can restore the maximum growth (closest to the wild-type growth rate) is designated as a candidate for the metabolic valve. The cut set is then identified as a possible candidate for having a branched network topology since it contains metabolic valve reaction. Its orthogonality score is then calculated. The scores calculated for the resulting network(s) can help confirm its orthogonality and eventually rank them as such. When the MCS algorithm is used in combination with the orthogonality score, the output is a strain metric that attempts to maximize growth-independent chemical production, and a set of genetic interventions required to establish it. This metric allows ranking the designs to determine the best combination of cut sets and metabolic valve reactions, which can then be systematically evaluated for the presence of a branched network topology as shown in Fig. 1. These sets are then manually curated to determine their suitability towards orthogonality and branched design.

**Thermodynamic and protein cost estimations.** We used the methodology described by Flamholz *et al.*[9] to calculate the thermodynamic driving force for each reaction and the corresponding protein costs. The protein cost of a reaction represents the amount of energy required to be expended by the cell such that a non-zero net flux is possible through the reaction.

The estimated cost accounts for the thermodynamics and the kinetics of the enzyme with respect to its interaction with the substrates and products of the reaction, and is also a function of the forward flux flowing through the reaction. We used the linear programming (LP) formulation[9] to estimate the minimum protein cost for every reaction in a network with physiological, thermodynamic and kinetic constraints on metabolite concentrations and reaction fluxes. A Michaelis–Menten kinetic rate law formulation was used to describe reaction fluxes using parameters $k_{cat}$ and $K_m$ obtained from *in vitro* enzyme assays data in the literature.

**Data availability.** The data that support the conclusions of this study are available from the authors on request.

**Code availability.** All relevant codes to run the algorithms are publicly available at http://www.labs.chem-eng.utoronto.ca/mahadevan/the-lab/downloads/

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

## Acknowledgements

The authors wish to acknowledge the following funding agencies and networks: Natural Sciences and Engineering Research Council of Canada; Genome Canada, Industrial Biocatalysis Network; Ontario Ministry of Research and Innovation; and BioFuelNet. The authors also wish to acknowledge Steffen Klamt for assistance with CellNetAnalyzer. The authors would also like to acknowledge Shawn Lu for help with the protein cost calculations.

## Author contributions

A.V.P. and R.M. conceived of the study. A.V.P. and S.S. designed the study, performed the simulations and developed the approach. A.V.P., S.S. and R.M. analysed the data, interpreted the results and wrote the paper.

## Additional information

**Competing interests**: The authors declare no competing financial interests.

**Publisher's note**: 

