## [Peer review file · Nature Communications]

Reviewers' comments:

Reviewer #1 (Remarks to the Author):

The manuscript by Pandit et al. provides a very exciting perspective into the critical problem of designing pathways and strains for metabolic engineering. It introduces various concepts, such as the pathway orthogonality that examines the interdependence of a pathway leading to chemical production with biomass generation, as well as the concept of a metabolic valve that allows carbon to be redirected from biomass to product formation. In addition, it offers the ability to examine the role of orthogonality in selecting substrates for producing a chemical and in the end it offers the ability to create design principles for more efficient chemical production.

The authors have demonstrated the applicability of this approach in the case of succinic acid production and have explored the possibility of using different carbon substrates such as xylose, instead of glucose.

The manuscript is well written and can be followed by someone that is a nonexpert in the field. Some minor edits are suggested:

1. In Figure 1, the authors need to define what is green (product EFMs) and blue (biomass EFMs) in the (B) section where the EFM matrix appears.
2. In page 6, in the third paragraph, I get the impression that the valve the authors are proposing consists of 6 gene targets, in order to minimize carbon loss with only five biomass precursors generated. This is a bit confusing as I was under the impression that the metabolic valve consists of a single gene.
3. In the concept of using dynamic control of metabolic fluxes, the authors should mention the possibility of using metabolic sensors for dynamic regulating fluxes. Such experimental approaches have been presented, such as for example in:

Improving fatty acids production by engineering dynamic pathway regulation and metabolic control.

Xu P, Li L, Zhang F, Stephanopoulos G, Koffas M.
Proc Natl Acad Sci U S A. 2014 Aug 5;111(31):11299-304.

Reviewer #2 (Remarks to the Author):

The paper describes a strategy for the biological production of chemicals based on uncoupling the production pathway from biomass production. This differs from the commonly suggested metabolic engineering approach of growth-coupling production. The authors define a metric called the “orthogonality score” that can be used to quantify the degree of uncoupling between a given production pathway and the cell’s growth pathways. The orthogonality score is computationally tested on several native and synthetic pathways showing that it can be used to select a suitable substrate, a good synthetic pathway, and identify metabolic valves that could be controlled to further uncouple the pathway from the growth metabolism. Thus the presented method could have useful applications for dynamic metabolic engineering. The manuscript is written clearly and the figures are also clear.

Major comments:

- 1) The approach presented in this manuscript is novel and interesting, but it is only demonstrated theoretically for one product. In order to be publishable in a journal of general interest I would expect either 1) experimental demonstration of the strategy or 2) computational demonstration of the strategy for a broader range of native and non-native chemical products. The latter is probably more doable in a short time frame.
- 2) The authors describe in the introduction growth coupling as the primary strategy for metabolic engineering. This is true in the world of computational strain design, but the majority of real world experimentally implemented metabolic engineering strategies are not growth coupled. In contrast, many of them are somewhat orthogonal to growth in the sense that much of the production happens in the non-exponential phase of fed-batch fermentation. The authors should qualify their statement on growth coupling as the primary strategy used in metabolic engineering.
- 3) The authors suggest that the uncoupling strategy presented in the manuscript would benefit from introducing a new alternative growth substrate. The authors analyze the pros and cons of different substrate utilization pathways extensively, but do not actually provide evidence that they are economically feasible in terms of the cost of substrate. It would be good to see a discussion of the basic techno-economics (substrate cost, product yield) of the proposed orthogonal pathways.

Minor comments:

- P4-L149: S_x should be clearly defined in the main text or in the figure text of fig. 1
- P5-L186: Orthogonal to what?
- P5-L187-189: Wouldn't a branched structure imply orthogonality, c.f. the earlier definition?
- P5-L192: What is wrong with valves linked to nutrient uptake?
- P6-L231: If there are several growth-independent succinate-producing EFM's it will not result in a single point, but rather a subspace on the growth=0 hyperplane

Figure3: The y-axis labels should be “Succinate yield (relative to theoretical maximum)” or something similar

Reviewer #3 (Remarks to the Author):

The manuscript by Pandit et al. describes a computational approach based upon metabolic modeling to suggest design strategies for chemical production that reduce interactions with native pathways. The main contributions are the development of an orthogonality metric and implementation of the approach to study the production of succinic acid in *Escherichia coli*. While the broad idea is conceptually interesting and potentially impactful, there are concerns about both the novelty and findings of this approach.

Comments:

1) Orthogonality concept/Figure 1 – The central concept of the study is the idea of an orthogonal pathway that has minimal interaction with other native network components. This is illustrated in Figure 1 and referenced numerous times throughout the manuscript. All of the presented illustrations for an orthogonal pathway (branched pathways, non-native heterologous pathways) are routinely used by the metabolic engineering field and thus this is more of a terminology change than a conceptual change.

2) Metabolic valve – The approach of identifying a metabolic valve to target as a control point for switching between growth and chemical production is an approach that has been acknowledged and experimentally implemented in a number of cases (Examples: PMID 25049420 and 22446695). The approach is commonly termed dynamic sensing or dynamic regulation and has been experimentally implemented to address the same concept presented here, the tradeoff that occurs between biomass production and chemical production.

3) Value of orthogonality metric – Overall, it is unclear if the orthogonality metric is a useful metric for evaluating designs. First, on Lines 129-130 the authors suggest orthogonality scores of “greater than ~0.5” are meaningful for determining pathway orthogonality. In subsequent results of the orthogonality metric, the authors have orthogonality scores of 0.41 (line 193) and a maximum of 0.43 (line 197). The only higher scores that are achieved are using synthetic, non-native heterologous pathways which by their nature must have higher orthogonality. Furthermore, given that this study is solely on one test case (succinic acid production), it is doubtful that this single example is sufficient to make any generalizable conclusions on the value of the orthogonality metric.

4) Minimum cut set approach – Why did the authors choose to use the minimum cut set approach to determine network states that optimize product yield with zero growth (Lines 158-223). This could have been accomplished as well by choosing an objective function defined by the chemical of interest and completing an optimization for maximum production.

5) The section on alternative substrate utilization pathways (Lines 235-264) are likely the most interesting and significant of the study. Superficially, the idea of starting from a niche (non-standard) input substrate and implementing a non-native pathway to use that substrate is intuitive. The value here is really in being able to do this analysis systematically for any chemical product of interest. While the results suggesting use of xylose or ethylene glycol as starting points are of interest, again the question is how amenable is this approach to analysis of additional targets outside of this one case?

6) Context – The choice of E. coli (using a core model) and succinic acid for illustrating the developed approach are curious. The use of a core model may restrict the ability to find alternative orthogonal pathways, and succinic acid is a core central metabolism component so it is unclear what the authors hoped to find in terms of orthogonal pathways. Thus, all of the results are not surprising given these constraints – there are no native orthogonal pathways and the only orthogonal pathways are ones that here non-native, heterologous pathways.

The authors also state that the type of orthogonality found here is not found in natural metabolism. However, there are a number of organisms that do naturally undergo biphasic growth that falls conceptually along the lines of what is presented here (e.g. *Clostridium acetobutylicum*).

7) Method – A key aspect of the calculation of the orthogonality metric is the binning of results into biomass or chemical production categories (S_x and S_t for subspaces and e_j^x and e_j^t for elementary flux modes). How is this determination made?

Minor comments:

1) Line 91 – The authors state that a core model of *E. coli* was used for simulations. Why was only a core model of *E. coli* used instead of any of the numerous more expansive *E. coli* models? There also is not an included description of the contents of the core model used.

2) Writing style – It is suggested that segments of the manuscript be re-written using terminology that is more accessible to a broad audience rather than referring to terminology relevant to people specific to this field (e.g. line 119, lines 144-148) where EFMs and subspace matrices S_t and S_x are used.

3) Table 2 – The values shown in Table 2 and in the text (lines 279 – 285) are inconsistent.

4) Figure 1 – The legend refers to green boxes. The only green boxes in the Figure are for the row in Figure 1B for EFM2 and around the y-axis on Figure 1C. Is that correct?

Reviewers' comments:

Reviewer #1 (Remarks to the Author):

The manuscript by Pandit et al. provides a very exciting perspective into the critical problem of designing pathways and strains for metabolic engineering. It introduces various concepts, such as the pathway orthogonality that examines the interdependence of a pathway leading to chemical production with biomass generation, as well as the concept of a metabolic valve that allows carbon to be redirected from biomass to product formation. In addition, it offers the ability to examine the role of orthogonality in selecting substrates for producing a chemical and in the end it offers the ability to create design principles for more efficient chemical production.

The authors have demonstrated the applicability of this approach in the case of succinic acid production and have explored the possibility of using different carbon substrates such as xylose, instead of glucose.

The manuscript is well written and can be followed by someone that is a nonexpert in the field. Some minor edits are suggested:

1. In Figure 1, the authors need to define what is green (product EFMs) and blue (biomass EFMs) in the (B) section where the EFM matrix appears.

We thank the reviewer for his comments. These changes have been made in the figure captions (L639-641 and L635-636) as well as main text L145-146.

2. In page 6, in the third paragraph, I get the impression that the valve the authors are proposing consists of 6 gene targets, in order to minimize carbon loss with only five biomass precursors generated. This is a bit confusing as I was under the impression that the metabolic valve consists of a single gene.

We have clarified this statement. The intervention set consists of 5 genes (plus three fermentative genes that are always blocked). However only one reaction of the five is a metabolic valve. Hence, in the example, five genes are targeted to *design* for orthogonality, with the fifth serving as a metabolic valve. Without the fifth gene being turned *off*, the cell cannot switch between growth and production modes. However, we do understand the reviewers concern about this being unclear. We have amended the main text to hopefully be clearer in L217-225.

The following text was corrected:

“Consider one sample strategy from that set of 131, which requires disruption to the five genes in addition to the standard fermentative enzymes (ldh, adh, ackr). These five consist of the gluconeogenic enzymes phosphoenolpyruvate synthase, both malic enzymes as well as succinyl-coa ligase (Fig. 2b). Under this strategy these four gene deletions are required to provide a singular pathway towards satisfying growth precursor requirements. The fifth genetic intervention, phosphoenolpyruvate carboxykinase is the metabolic valve that can be manipulated to direct flux towards biomass to chemical synthesis. If the metabolic valve is off, only five biomass precursors that also belong to the succinic acid biosynthesis pathway can be synthesized, and the mean orthogonality score is 0.55, a relatively high

value. Hence this combination of genetic interventions allow the metabolism to be recast into a design with one metabolic valve.”

3. In the concept of using dynamic control of metabolic fluxes, the authors should mention the possibility of using metabolic sensors for dynamic regulating fluxes. Such experimental approaches have been presented, such as for example in:

The reviewer is correct to point to the idea of using metabolic valves to control flux have been used in the literature. While we do cite these examples in the work by way of review papers, the reviewer is correct to point that we should highlight these experimental approaches in more detail. These changes have been made to the manuscript in L363-365. We have referenced a total of six experimental approaches including the ones the reviewer suggested. (New citations added: PMID 25049420, 19558964, 23026120, 24576819, 22446695, and 25542851).

Improving fatty acids production by engineering dynamic pathway regulation and metabolic control.
Xu P, Li L, Zhang F, Stephanopoulos G, Koffas M.
Proc Natl Acad Sci U S A. 2014 Aug 5;111(31):11299-304.

Reviewer #2 (Remarks to the Author):

The paper describes a strategy for the biological production of chemicals based on uncoupling the production pathway from biomass production. This differs from the commonly suggested metabolic engineering approach of growth-coupling production. The authors define a metric called the “orthogonality score” that can be used to quantify the degree of uncoupling between a given production pathway and the cell’s growth pathways. The orthogonality score is computationally tested on several native and synthetic pathways showing that it can be used to select a suitable substrate, a good synthetic pathway, and identify metabolic valves that could be controlled to further uncouple the pathway from the growth metabolism. Thus the presented method could have useful applications for dynamic metabolic engineering. The manuscript is written clearly and the figures are also clear.

Major comments:

1) The approach presented in this manuscript is novel and interesting, but it is only demonstrated theoretically for one product. In order to be publishable in a journal of general interest I would expect either 1) experimental demonstration of the strategy or 2) computational demonstration of the strategy for a broader range of native and non-native chemical products. The latter is probably more doable in a short time frame.

We thank the reviewer for their comment. While we agree that experimental demonstration of the strategies proposed would make for a stronger paper, we believe that this is outside the scope of the current publication. As a side, we are currently in the process of experimental demonstrating these concepts using orthogonal substrates and have obtained some initial very preliminary results that require additional validation and testing. In regards to the current publication, while we do present additional computational demonstration of the strategy in the Supplementary Information, we do agree with the reviewer that having a broad range of native and non-native chemical products would make for a stronger

publication. As a result, we have expanded the analysis in our supplementary Information to cover a broader range of products and included some of this analysis in the main text. While we feel we cannot include all the detailed analysis for all products shown in the supplementary information in the central text, we hope that the additional analysis included in the supplementary should address the reviewer's concern and demonstrate generality in our approach.

2) The authors describe in the introduction growth coupling as the primary strategy for metabolic engineering. This is true in the world of computational strain design, but the majority of real world experimentally implemented metabolic engineering strategies are not growth coupled. In contrast, many of them are somewhat orthogonal to growth in the sense that much of the production happens in the non-exponential phase of fed-batch fermentation. The authors should qualify their statement on growth coupling as the primary strategy used in metabolic engineering.

The reviewer is correct to point out that real world implementation of metabolic engineering strategies is often not growth coupling. Although, it often is difficult to achieve complete growth independent production as well. Indeed, this has been a central motivation of our work and while we alluded to this idea, it was not sufficiently clear that this was our intention. Hence, given the reviewer's comment we believe that it is important to qualify our statements on growth coupling more explicitly. We have made these changes to the manuscript in L66-67 as well as L374-377 and hope the reviewer is satisfied with this revision.

3) The authors suggest that the uncoupling strategy presented in the manuscript would benefit from introducing a new alternative growth substrate. The authors analyze the pros and cons of different substrate utilization pathways extensively, but do not actually provide evidence that they are economically feasible in terms of the cost of substrate. It would be good to see a discussion of the basic techno-economics (substrate cost, product yield) of the proposed orthogonal pathways.

We thank the reviewer for this comment. Indeed, the economics of utilizing a substrate are central to the industrial scale-up of microbes that use non-native substrates or non-native pathways. However, while we believe that a full techno-economic analysis of these pathways fall largely outside the scope of the present study, we have included a short analysis of substrate cost and yield to pyruvate as a comparison standard in the Supplementary Information. Given that pyruvate is a central building block molecule, we believe it provides a useful substitute for a host of chemicals. We hope the reviewer will find this analysis sufficient.

Minor comments:

P4-L149: S_x should be clearly defined in the main text or in the figure text of fig. 1

Changes made in L146-157.

P5-L186: Orthogonal to what?

Changes made in L184 - L186.

P5-L187-189: Wouldn't a branched structure imply orthogonality, c.f. the earlier definition?

We would like to thank the reviewer for this comment. We have clarified that statement. We meant to explore the idea of whether a non-orthogonal structure characterized by a low score could be modified by gene deletions to raise the score and arrive at a new network layout resembling a branched structure.

P5-L192: What is wrong with valves linked to nutrient uptake?

Nutrient based valves can include oxygen, nitrogen and phosphate limitations. While oxygen based nutrient valves have been observed in large scale bioprocesses (e.g. succinic acid production from glucose is a two stage fermentation), and nitrogen limitation is used to produce citric acid, we ignored these types of valves because they are largely understood. The recent focus in metabolic engineering has been the design and use of complex synthetic circuits to control gene expression (e.g. via a synthetic toggle switch). In light of these approaches, our work has been to understand how reworking the design of the central metabolism may allow the simplification of these circuits so that rather than employing a multi-gene control, it may be possible to achieve desired production targets by manipulating a single gene. We have now included this explanation in the discussion text L378-390. Of course, it is conceivable that these gene level valves could be combined with the valves related to nutrient uptake to provide an additional layer of flexibility in controlling metabolism.

P6-L231: If there are several growth-independent succinate-producing EFM's it will not result in a single point, but rather a subspace on the growth=0 hyperplane

The reviewer is correct to point out that the $\mu = 0$ hyperplane results can contain many EFMs. These points will vary in their product yield. However, the ValveFind algorithm attempts to reduce the elementary flux modes in the hyperplane to a single point if possible (depending on the threshold product yield demanded) and, if not, then to a small high yield region of the production envelope. This can be visualized in Figure 4. As the production envelopes shrink along the x-axis, the y-axis also shrinks indicating the removal of low-yielding EFMs along the $\mu = 0$ hyperplane. We have now included this explanation of how the ValveFind algorithm works in the methods section L461-470.

Figure3: The y-axis labels should be "Succinate yield (relative to theoretical maximum)" or something similar

Changes made.

Reviewer #3 (Remarks to the Author):

The manuscript by Pandit et al. describes a computational approach based upon metabolic modeling to suggest design strategies for chemical production that reduce interactions with native pathways. The main contributions are the development of an orthogonality metric and implementation of the approach to study the production of succinic acid in Escherichia coli. While the broad idea is conceptually interesting and potentially impactful, there are concerns about both the novelty and findings of this approach.

We thank the reviewer for the positive comments and suggestions for improvements.

Comments:

1) Orthogonality concept/Figure 1 – The central concept of the study is the idea of an orthogonal pathway

that has minimal interaction with other native network components. This is illustrated in Figure 1 and referenced numerous times throughout the manuscript. All of the presented illustrations for an orthogonal pathway (branched pathways, non-native heterologous pathways) are routinely used by the metabolic engineering field and thus this is more of a terminology change than a conceptual change.

We agree with the reviewer that the metabolic engineering field has used the terms branched and non-native heterologous pathways, we do not agree that our use of orthogonal pathways is a terminology change rather than a conceptual change. Branched and non-native pathways are routinely engineered into strains however, we have not found in the literature any work that assesses the degree of cellular interactions experienced by these pathways. Moreover, a non-native heterologous pathway need not be orthogonal based on our metric, given that there are potential interactions with the cellular metabolism through co-factors and shared intermediates that will be explicitly minimized using our definition of the metric. Therefore, we believe that our definition of the orthogonality metric explicitly quantifies the nature of these interactions of the product producing pathway with the native metabolism and hence is distinct from previously used concepts of branched and non-native heterologous pathways that we believe are ad-hoc and not systematic and quantifiable like the concepts in our paper. In lines 315-320 we describe that orthogonality is closely related to modularity however, a fundamental difference is in the nature by which orthogonality is identified and characterized by quantifiable metric.

2) Metabolic valve – The approach of identifying a metabolic valve to target as a control point for switching between growth and chemical production is an approach that has been acknowledged and experimentally implemented in a number of cases (Examples: PMID 25049420 and 22446695). The approach is commonly termed dynamic sensing or dynamic regulation and has been experimentally implemented to address the same concept presented here, the tradeoff that occurs between biomass production and chemical production.

The reviewer is correct to point out that there have been experimentally implemented changes between switching between growth and chemical production. While we do refer to experimental work via review articles, we have expanded the text to more explicitly acknowledge these works. (New citations added: PMID 25049420, 19558964, 23026120, 24576819, 22446695, and 25542851).

However we would like to mention that our work has two additional novel contributions that differ from these studies. First, we address the issue of selecting metabolic valves using a systematic design rather than an arbitrary one. The focus of this work is more on addressing the structural properties arising from elementary flux modes and the advantages these structures confer as opposed looking at it from just a growth-coupled or growth vs production trade-offs. To our knowledge, this is the first study that provides a mathematical framework for designing such valves. To that end, we have included a case study in Supplementary Information on how valve selection can affect cell wide interactions on globally regulated metabolites pools such as NADPH and NADH. Secondly, while dynamic sensing or dynamic regulation has been shown experimentally, its use has been limited to simple systems. Our present work addresses the design of metabolic strategies with the goal of establishing viable metabolic valves by reworking substrate utilization pathways. These strategies arise from core design principle rather than as an ad hoc consideration of growth coupled designs.

3) Value of orthogonality metric – Overall, it is unclear if the orthogonality metric is a useful metric for evaluating designs. First, on Lines 129-130 the authors suggest orthogonality scores of “greater than ~0.5” are meaningful for determining pathway orthogonality. In subsequent results of the orthogonality metric, the authors have orthogonality scores of 0.41 (line 193) and a maximum of 0.43 (line 197). The only higher scores that are achieved are using synthetic, non-native heterologous pathways which by their nature must have higher orthogonality. Furthermore, given that this study is solely on one test case (succinic acid production), it is doubtful that this single example is sufficient to make any generalizable conclusions on the value of the orthogonality metric.

We thank the reviewer for this comment. As with a similar comment from another reviewer, we have expanded analysis in the Supplementary Information that considers many more substrate product pairs. Moreover, we have added additional text to Results and Discussion that comment on these findings. We hope the reviewer will find these examples more useful.

With respect to line 193 and 197, these values 0.41 and 0.43 are in line with our analysis that orthogonality scores greater than 0.5 are “orthogonal” networks. 0.41 and 0.43 were scores for natural metabolism – consistent with non-orthogonal networks. Finally, as stated earlier, all non-native heterologous pathways need not be orthogonal as some of them can potentially share intermediates and reactions that decrease their orthogonality score and hence, a metric is valuable to understand these interactions in a systematic manner. We provide an example of ethylene glycol metabolism through three different routes of assimilation all of which have very wide ranging scores. The purpose of the orthogonality metric is not to be all and end all for analyzing pathways, but rather its usefulness lies in that it provides a quantitative, unbiased metric based on network wide interactions for understanding that engineering substrate utilization pathways is an important consideration of the metabolic engineering process that is by-and-large currently overlooked.

4) Minimum cut set approach – Why did the authors choose to use the minimum cut set approach to determine network states that optimize product yield with zero growth (Lines 158-223). This could have been accomplished as well by choosing an objective function defined by the chemical of interest and completing an optimization for maximum production.

To explain why we used a minimum cut set approach and not flux balance analysis that optimizes the product yield, we need to explain the differences between these two modelling techniques. Minimal cut sets are solutions to the dual problem of constraining the elementary flux modes of a metabolic network to a desired region. EFMs are the steady state solution of the metabolic network and they represent the simplest biochemically meaningful flux vector possible. When the reactions are irreversible, they are the generating vectors of the flux cone and any solution within the cone is a linear combination of those generating vectors. Therefore, identifying the minimal cut sets creates a solution space in the flux cone with the desired product yield.

In the minimal cut sets approach, the goal is to eliminate the solution space containing low producing metabolic modes through gene deletions and hence guaranteeing the production above a threshold regardless of the objective function in the constraint-based modeling framework. Hence, any feasible solution represents a potential strain design strategy in contrast to the FBA based solution. This subtle idea is an important strength of the minimal cut sets that is often underappreciated and hence, we would like to reiterate the strength of the cut set based approach relative to the FBA based methods.

In contrast, Flux balance analysis identifies flux vectors in the null space that maximizes a desired objective function. The most typical objective being maximized is the growth rate, or the biomass equation. Interestingly, this FBA solution corresponds to one EFM under the minimal flux condition, hence there exists a relationship between flux balance analysis and elementary flux modes. The single solution obtained by choosing an objective function defined by the chemical of interest and completing an optimization for maximum production as defined by an OptKnock solution also corresponds to an elementary flux mode.

Why don't we choose the chemical production objective to determine the network states and calculate orthogonality? To answer that question we have to recognize that the FBA solution to the objective function is a single elementary flux mode. Hence, comparing the solution of the biomass objective function and the chemical production objective function results in a set of reactions that are different between these two solutions that need to be turned on and off to rewire the cell completely from a growth to a production state. It is really important recognize that there may be hundreds of reactions that are different across these two modes. For example, using the *E. coli* (core) model and optimizing for growth results in 48 non-zero reactions where as optimizing for succinic acid results in 37 non-zero reactions with 25 reactions that change flux among them. Hence, it would be impractical to regulate so many reactions to switch between these states.

However, what we are interested in instead is something slightly different. We want to identify a minimal set of genetic interventions, ideally a single reaction if possible, that can shift the production of the cell from one state (growth) to another (chemical production) and measure the average interactions within this shift. This approach inherently requires considering a solution space (of several possible solutions) rather than a single solution vector (guaranteeing a minimal chemical production rate and a solution space corresponding to a growth rate greater than a minimal threshold). Hence we are required to consider a set of elementary modes in addition to the two vectors that are solutions to an FBA objective function of the wild-type network. The reactions contained in these sets then become targets for manipulation.

5) The section on alternative substrate utilization pathways (Lines 235-264) are likely the most interesting and significant of the study. Superficially, the idea of starting from a niche (non-standard) input substrate and implementing a non-native pathway to use that substrate is intuitive. The value here is really in being able to do this analysis systematically for any chemical product of interest. While the results suggesting use of xylose or ethylene glycol as starting points are of interest, again the question is how amenable is this approach to analysis of additional targets outside of this one case?

We thank the reviewer for their positive comments on the alternative substrate utilization pathways. We would argue that non-standard input substrate is not that intuitive. The literature really doesn't show too many examples of non-standard substrates and non-standard pathways. Only a handful of studies which have been cited in the manuscript, to our knowledge exist, most of which were published during the course of the preparation of this manuscript. However, while we have provided a more detailed examination of targets outside this case in the Supplementary Information we have provided additional references to it and expanded its analysis in the main text to make it more obvious to refer to. We hope this strengthens this component of the paper.

6) Context – The choice of *E. coli* (using a core model) and succinic acid for illustrating the developed approach are curious. The use of a core model may restrict the ability to find alternative orthogonal pathways, and succinic acid is a core central metabolism component so it is unclear what the authors

hoped to find in terms of orthogonal pathways. Thus, all of the results are not surprising given these constraints – there are no native orthogonal pathways and the only orthogonal pathways are ones that here non-native, heterologous pathways.

One of the reasons we use the core model, as opposed to the genome scale model is because the analysis we undertake is computationally intractable for genome scale models as the EFM calculations require a substantial amount of memory and computational time. However, we would make the case that a core model provides sufficient detail as to be useful for drawing lessons from and making conclusions about the structure of metabolism. In particular it is worth noting that secondary metabolism is derived from the core metabolism and its metabolites, and the various subsystems in the secondary metabolism tend to be orthogonal and modular in nature. Hence, if the primary metabolic pathway structures do not lend themselves to orthogonality, minimal interactions and pathway independence, then secondary metabolism cannot either.

In response to the second comment, in reality, natural pathways that are indeed orthogonal exist. We provide an example of that in the supplementary information and provide additional references to it in the main text when we described xylose metabolism which is natural but orthogonal to *E. coli*. The natural glycerol metabolism also exhibits a higher degree of orthogonality for 2,3 butanediol production but not for other chemicals relative to other substrate pairs. In this regard, the orthogonality is determined by the chemical and the substrate pair and the pathway used for their interconversion rather than whether they are native or non-native. However, while this information was included in the supplementary and in light of the reviewers other comments, we have provided more analysis in the results and the supplementary. We hope that the reviewer will be satisfied with this additional analysis. Finally, we have also included an additional case study on the conversion of glycerol to the compounds 1,3-propanediol and 3-hydroxypropionic acid in the supplementary.

The authors also state that the type of orthogonality found here is not found in natural metabolism. However, there are a number of organisms that do naturally undergo biphasic growth that falls conceptually along the lines of what is presented here (e.g. *Clostridium acetobutylicum*).

We thank the reviewer for this comment and agree that there are in fact many organisms that can undergo a biphasic response. *Clostridium* species can either perform acidogenesis or solventogenesis depending on their environmental conditions or presence of stress, but there are other examples in nature that also allow for a rapid change in phenotype (diauxic growth, apoptosis, lambda phage switch). So in light of these counter-examples, let us try to clarify our original statement pertaining to the idea that orthogonality is not found in natural metabolism.

We believe there are three questions at the heart of the reviewer's comment. First, if natural metabolism is capable of biphasic growth, does this not mean that natural metabolism is orthogonal in the sense that we describe? Secondly, can non-orthogonal networks exhibit complex, biphasic shifts? And thirdly, does the mathematical definition of orthogonality fall apart because it does not describe the natural phenomena of biphasic growth which does not require a synthetic central metabolism to shift between phenotypes?

The first two questions are related. To answer the first question we refer to our original definition and purpose of the orthogonality score which was, quantifying the ability of a cellular network to function independently at performing two tasks by calculating a metric that quantifies this independence based on the shared elements (reactions) between these two tasks. The orthogonality score, by itself, says nothing about whether metabolism can switch between two points (efms), but only whether the number of reactions that need to be turned off or on, to do so, are many or few. Therefore, when we describe that

natural metabolism is not orthogonal, we do so in the context of the orthogonality score and not based on switching – which is something we expand only in a later section. Since natural metabolism does not have an orthogonality score of 1, it cannot be orthogonal. Moreover, since the orthogonality score for natural metabolism is less than for the synthetic pathways, it is less orthogonal than synthetic pathways. Therefore, irrespective of metabolism being capable of biphasic growth or not, the structure of the metabolism itself, based on shared similarities between two different tasks, is not orthogonal.

With respect to the second question, does this not mean that natural networks cannot exhibit complex changes in metabolism because of their low score? The answer to this yes they can. However, it's important to qualify that 'yes'. We have to recognize that the changes that occur in the metabolism are a result of a naturally programmed response in the cell that is controlled and arrived at in an evolutionary context. There are many layers of regulation guiding these shifts that include transcription factors and protein level interactions working simultaneously and in a coordinated manner that is perhaps difficult to replicate synthetically except in the simplest of cases. One of our concerns in this paper is the application of using dynamic strategies for metabolic engineering purposes. We accept the premise that it is possible to control the expression of many genes, transcription factors and regulatory elements in the cell akin to engineering a biphasic shift from growth to production. However, we challenge the ease and skill with which this can practically be done. Hence, the ValveFind and Orthogonality Score provide a rational basis to simplify metabolic engineering and guide the design of synthetic metabolic networks to achieve those ends by providing a simpler alternative to the control of metabolism.

Thirdly, does our mathematical definition of orthogonality fall apart because it does not account for this complex behaviour seen in natural metabolism? We argue once again that it does not. Recognize that in metabolic engineering, we attempt to repurpose the cell towards producing a non-natural chemical production objective. To accomplish this task, the process often requires removing as much of the regulation as possible that exists in the cell to constrain chemical production. Dynamic metabolic engineering requires an efficient method to switch between tasks (or objectives). The mathematical definition of orthogonality based on the elementary flux modes permits that because it captures the degree to which parts of one metabolism impact another.

Finally, our original claim was made in the section titled “Natural metabolism is not orthogonal.” The orthogonality score measured the degree to which elementary flux modes of the cell exhibit similar or different characteristics based on their production objectives. In fact, *Clostridium* cells that undergo acidogenesis instead of solventogenesis do exhibit marked differences in the profiles of their metabolism. For example, while acidogenesis permits the use of pentose phosphate pathway reactions in various different elementary flux modes, no elementary flux mode exists for solventogenesis that has an active pentose phosphate pathway. At the same time, as might be expected, the orthogonality score for this natural system is quite low when we analyze the EFMs determined Kumar *et al* (PMID: 24852622), for the *Clostridium* core model. The score is 0.33 but this example, as in many in nature is an exception to how complex natural systems have evolved and adapted regulatory solutions to address complex, bi-phasic behaviour that requires re-routing of cell fluxes.

7) Method – A key aspect of the calculation of the orthogonality metric is the binning of results into biomass or chemical production categories (S_x and S_t for subspaces and e_j^x and e_j^t for elementary flux modes). How is this determination made?

S_x and S_t are determined by whether e_j^x or e_j^t contain a non-zero flux through v_x or v_t . We have expanded the methods to more clearly explain this methodology.

Minor comments:

1) Line 91 – The authors state that a core model of *E. coli* was used for simulations. Why was only a core model of *E. coli* used instead of any of the numerous more expansive *E. coli* models? There also is not an included description of the contents of the core model used.

We thank the reviewer for this comment and we have revised the manuscript Methods section to explain this decision. The computation of elementary flux modes is a computationally burdensome endeavour that requires significant computational time and memory. As a result, it is currently infeasible to compute elementary flux modes for genome scale models. However, given that secondary metabolism is derived from the central metabolism by modular, and largely independent subsystems (e.g. amino acid metabolism), we believe that analysis using a core model provides a sufficiently detailed framework to draw conclusions upon.

A more detailed description of the *E. coli* model (Orth *et al.*, 2010) as well as the modifications made to the model for the other pathways is included in the Supplementary Information V. We hope these changes address the reviewers concerns.

2) Writing style – It is suggested that segments of the manuscript be re-written using terminology that is more accessible to a broad audience rather than referring to terminology relevant to people specific to this field (e.g. line 119, lines 144-148) where EFMs and subspace matrices S_t and S_x are used.

Point taken. We agree we can make this more accessible and have modified these lines to make it text more accessible to a broader audience where possible.

3) Table 2 – The values shown in Table 2 and in the text (lines 279 – 285) are inconsistent.

Thank-you. We have corrected this error.

4) Figure 1 – The legend refers to green boxes. The only green boxes in the Figure are for the row in Figure 1B for EFM2 and around the y-axis on Figure 1C. Is that correct?

Yes, that is correct – it is meant to refer to the subspace S_x . However, due to some misunderstanding regarding the caption of Figure 1, we have revised it to hopefully ensure that it is clearer.

REVIEWERS' COMMENTS:

Reviewer #1 (Remarks to the Author):

The revised manuscript looks good for publication. I have no additional comments

Reviewer #2 (Remarks to the Author):

The authors have addressed my comments comprehensively. The manuscript is now clearer and makes a good case for the proposed method to be a valuable tool in metabolic engineering.

One minor comment:

Line 66-69. Claiming that growth coupling is totally incongruent with real-world implementations is a bit harsh. There are fermentations that produce essentially growth coupled products like ethanol in yeast, but this is usually not the case.

Response to Reviewers

Comment: Line 66-69. Claiming that growth coupling is totally incongruent with real-world implementations is a bit harsh. There are fermentations that produce essentially growth coupled products like ethanol in yeast, but this is usually not the case.

Response: Specifically, we modified the sentence on the incongruence of the growth-coupled methods as follows:

“Finally, we believe that the approach provides a new paradigm of metabolic engineering strategies for chemicals, in contrast to the existing growth coupled strategies which can be difficult to implement in practice.”